# Pericardial Adhesion and Chronic Non-Specific Neck Pain following Thoracentesis: An Osteopathic Approach

**Bruno Bordoni** [1,*] **and Allan Escher** [2]

1   Department of Cardiology, Foundation Don Carlo Gnocchi Istituto di Ricovero e Cura a Carattere Scientifico (IRCCS), Institute of Hospitalization and Care, S Maria Nascente, Via Capecelatro 66, 20100 Milan, Italy
2   Anesthesiology/Pain Medicine, H. Lee Moffitt Cancer Center and Research Institute, 12902 USF Magnolia Drive, Tampa, FL 33612, USA; allan.escher@moffitt.org
*   Correspondence: bbordoni@dongnocchi.it

**Abstract:** Cardiovascular diseases (CVDs) are the leading cause of death globally. Morbidity and disability related to non-fatal events are increasing exponentially. There are several symptoms that may arise after invasive therapeutic approaches such as coronary artery bypass graft (CABG), including chronic pain in anatomical areas connected to the mediastinum; these pains can be found not only initially after surgery but also years later. We present a case where non-specific neck pain (NNP), in a patient undergoing CABG five years earlier, was resolved with an osteopathic technique, working the pericardial area. To the knowledge of the authors, it is the first article illustrating an osteopathic approach with resolution of NNP, with a manual technique used on the pericardial area.

**Keywords:** osteopathy; osteopathic manipulation; fascia; non-specific neck pain; diaphragm

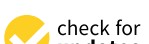



## 1. Introduction

Acute myocardial infarction (AMI) caused by coronary heart disease (CHD) is the result of one or more coronary artery stenoses [1]. AMI is included in the category of cardiovascular diseases (CVDs), which are the leading cause of death in the population, also increasing morbidity and disability [2]. The stenosis causes ischemia and necrosis of the myocardial tissue due to the absence of blood nourishment; the inflammatory response that occurs in the non-perfused area will generate fibrotic tissue and possible adhesions between the myocardium and the visceral pericardium and/or adhesions between the pericardial leaflets and the myocardium [1–4]. One of the elective invasive approaches to reperfuse tissue is the coronary artery bypass graft (CABG).

This practice, especially if performed with median sternotomy, ministernotomy or lateral minithoracotomy, stimulates the post-surgical formation of peritoneal adhesions [5–7]. CABG is not free from dangers and complications during and after surgery, which can range from surgical site infection to the appearance of arrhythmias, up to the event of a periprocedural AMI and pleural effusion [8]. The phrenic nerve can undergo a transient or permanent injury (up to a maximum of 75% of patients), with an ipsilateral elevation of the diaphragm, or, in the most severe cases, both domes are dysfunctional [9]. The intercostal nerves undergo a structural and functional alteration of the myelin and meningeal layers, and the appearance of fibrosis [10]. Another event that can slow down the post-surgical course is ipsilateral shoulder pain (ISP), a type of pain that can affect up to 97% of patients undergoing CABG [11]. Other areas of the body are tender after sternotomy, such as the neck area, causing the patient to take more pain relievers [12]. The case report presents a patient with chronic non-specific neck pain (NNP) in the presence of pericardial adherence following thoracentesis and prior CABG. To the knowledge of the authors, it is the first article illustrating an osteopathic approach with resolution of NNP, with a manual technique used on the pericardial area.

## 2. Case Presentation

A 36-year-old male patient with previous cardiac surgery in median sternotomy, not infarcted, was examined for NNP in the cervical area. Surgery was performed five years earlier, with double CABG (obtuse marginal branch1-posterior interventricular coronary artery-saphenous vein). Four days after surgery, the patient underwent thoracentesis due to a left pleural effusion involving the pleural space up to the sixth rib. The drugs currently being taken by the patient are aimed at blood pressure control (beta-blocker and angiotensin-converting enzyme—ACE—inhibitor), and are taken due to the presence of hypercholesterolemia and dyslipidemia (statins). The patient complained of pain and stiffness throughout the posterior cervical area, without irradiation, with movement limitations that increase with rotations and inclinations of the neck to the right.

The relief came from the extension, and from the opposite rotation and inclination. The deep pain was intermittent and present only with active and passive (performed by the operator) right movements. Once a day, the patient takes a pain reliever (non-steroidal anti-inflammatory drugs—NSAID). The patient did not report direct trauma or cervical pathologies or axial alterations with negative results through previous instrumental tests (magnetic resonance and X-rays). Blood tests do not highlight systemic diseases (infections, arthritis or autoimmune diseases, and metabolic/endocrine diseases).

Air saturation was 100%, and with the complete absence of dyspnea. With the visual analogue scale (VAS), if pain was solicited, the score was six; the neck disability index (NDI) reported a value of 24 (moderate disability). We excluded patent foramen ovale (PFO), which can be a source of neck pain, because it was not present during surgery. We excluded carotid diseases (stenosis, dissections), which can cause pain in the neck, thanks to the color Doppler ultrasound of the carotids previously performed, before the osteopathic visit. We excluded the temporomandibular joint (TMJ) as the origin of the symptoms, due to the absence of popping, clicking or crepitus, as well as the lack of pain and muscle tension on movement and palpation by the osteopath (pterygoid, temporal, and masseters), and no hearing disorders and/or alterations of the dental class. The patient denies bruxism or nocturnal snoring. We excluded Eagle's syndrome and hyoid bone syndrome due to the lack of related symptoms (dysphagia, odynophagia, etc.), and due to the palpation of the hyoid bone. We performed several manual differentiation tests (less specific like the O'Donoghue's test or more specific like the Sharp–Purser test), but always inconclusive. The cervical compression and distraction tests were also negative.

We manually evaluated the vertebrae that affected rotation and inclination the most, such as the first two cervical vertebrae (affect total rotation by about 63–73%), C3–C4 and C6–C7 (affect inclination by about 11.7–20.3%), respectively [13]. By isolating the movements of the individual facet joints, we did not find anomalies such as to explain the limitation of movements or any triggers for the onset of pain. Palpation of the cervical musculature and of the posterior cervicothoracic area revealed tissue suffering of the deep muscles but not of the more superficial muscles. Considering that scars may be a source of symptoms not necessarily at the site of the lesion, we evaluated the sternal scar and the area of the sixth posterolateral intercostal space (access area for thoracentesis), manually [14]. With small delicate pressures with an open palm on the scar area, we induced micro-movements (caudal, cranial, right, and left direction, oblique and rotational movements) to try to highlight possible adhesion brakes or possible pain. The induced micromovements were strongly limited in the directions followed, but without pain, indicating a possible presence of adhesions under the scar areas. Adherent lesions between the serous cavities and the lungs and/or pericardium following surgery are common [15]. To try to identify the probable source of neck movement limitation and pain, we performed a manual inhibition test.

This test consists of pressing a tissue area (sternum) while resting the other hand on the symptomatic area (the neck). Then, the clinician tries to perceive a change in tone. Finally, the cervical spine area is pressed, and the response is felt from the sternal area [16]. The tissue that does not show any tone variation is, very similarly, the area of symptomatological

derivation. In this case, the sternal and thoracentesis access areas were the areas where the tissue tone did not change. After performing the Bordoni diaphragmatic test (BDT), the outcome gave a positive result of dysfunction of the diaphragm muscle [17]. We decided to take a chest X-ray (Figure 1).

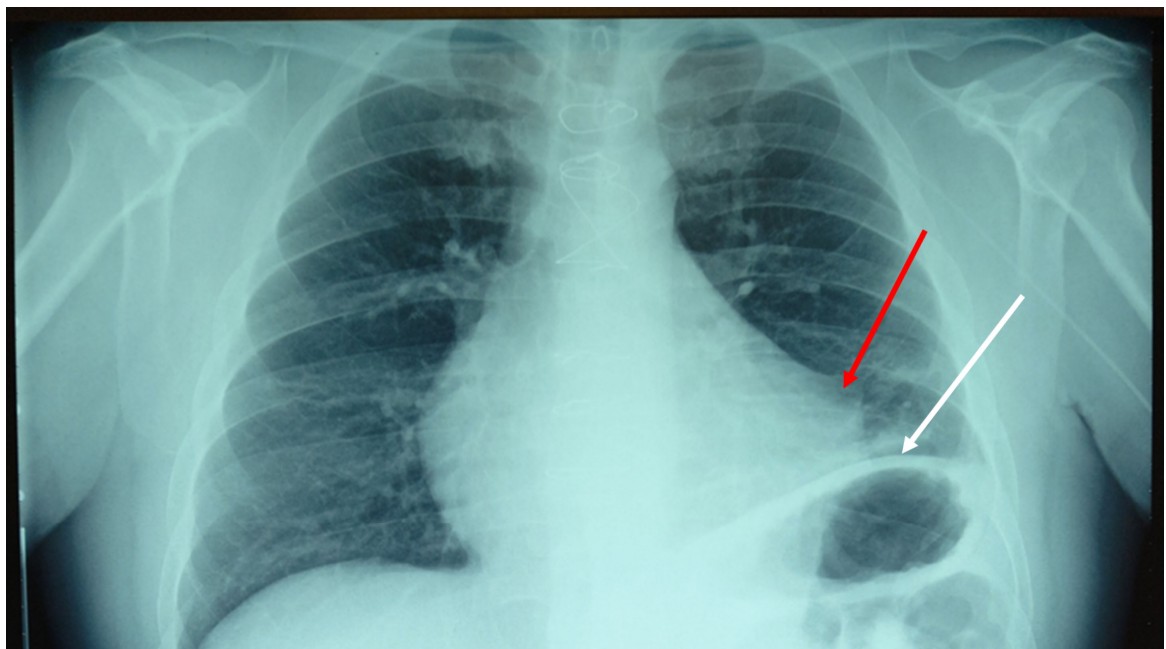

**Figure 1.** Anteroposterior chest X-ray shows an adhesion between the pericardial tissue and the pleural tissue of the left mediastinal area, probably due to a previous thoracentesis (red arrow). The left hemidiaphragm is visibly elevated due to a possible uninvestigated ipsilateral phrenic nerve lesion (white arrow).

The result of the instrumental examination highlighted an adherence between the pericardium and the pleura in the left mediastinal area, and an elevation of the left hemidiaphragm. We decided to perform an osteopathic treatment with a technique used in a previous trial, that is, an indirect approach [18]. The goal was to reduce pain and improve movement function. The clinician places one palm on the sternal area (involving the anterior pericardial area) and the other hand, open posteriorly, on the vertebral area of T10–T12; the latter thoracic area corresponds to the posterior attachment of the pericardium [19].

With this manual support to the chest, we tried to "enclose" the pericardial area. The clinician does not induce any movement but follows the movement of the tissues underlying the hands; when the perceived tissue displacement is homogeneous in amplitude, traction force, and times of expression, the technique can be said to be finished. The patient remains seated, while the operator is seated next to the patient. The application time can last several minutes. At the end of the osteopathic technique, when reevaluating the patient, the active and passive movements of the neck had no limitations; utilizing the previous assessment scales (VAS and NDI), the patient did not complain of any pain. After a week, the patient (contacted by telephone) described the recurrence of pain but with a much-reduced entity compared to the period preceding the osteopathic treatment and without taking NSAIDs.

Movements were unrestricted. We decided to see the patient again (one week after the first treatment), using the same osteopathic approach. After two weeks, the perceived pain was further decreased, but constantly present, and the neck movements improved more. After 3 months of treatment with the same approach, with one monthly session, there was no limitation of movements (rotation and tilt to the right); the pain returned two weeks after the osteopath visit, but the painful entity perceived by the patient had greatly decreased, with a VAS score of 1 and NDI with a value of 10 (minimum disability). We decided, in agreement with the patient, to have one session every six months, as maintenance therapy.

## 3. Discussion

There are many symptoms that can derive from adhesions, such as local and/or referred pain, or they can cause symptoms not necessarily related to pain, such as being the cause of constrictive pericarditis [20]. In this case report, a gentle osteopathic approach drastically reduced the symptoms of NNP in a patient with previous CABG and previous thoracentesis for left pleural effusion. The adhesion formed after thoracentesis united the left parietal pleura and the ipsilateral parietal pericardial area, forming nonphysiologically mechanical tractions. We can hypothesize that these new force vectors from the viscera may have conditioned the surrounding tissues, such as the diaphragm (through the lower triangular ligaments of the lung and the pericardial-diaphragmatic fascia), pulling the left hemi-cupola upwards. In the presence of adhesions, unfortunately, we do not have clear indications about the possible symptoms, and we do not have a gold standard of manual treatment. We know that the presence of pleural effusion pushes the corresponding portion of the diaphragm downwards, while thoracentesis helps diaphragmatic repositioning [21]. Probably, a possible concomitant lesion of the phrenic nerve (not evaluated), and the presence of visceral mediastinal adhesions, could have facilitated the cranial movement of the diaphragm muscle. The same adhesions present in the patient could have created traction towards the mediastinum of the connected fascial structures. The endothoracic fascia, which covers the pleurae and pericardium, derives from the deep fasciae of the neck, such as the investing, pretracheal, and prevertebral fascia. The investing fascia attaches to the nuchal ligament and mastoid process of the temporal bone [22]. The pretracheal layer involves the base of the skull up to the pericardium posteriorly, while the prevertebral fascia fuses with the vertebrae of the cervicothoracic tract, the nuchal ligament, and the deep musculature of the cervical vertebrae, as well as at the base of the skull and the Sibson's fascia (pleural dome) [22]. We can assume that the non-physiological force vectors generated by the visceral adhesion towards the mediastinum could be the cause of a mechanical imbalance expressed by the neck. In fact, the action of bringing the neck to the right (inclination and rotation) created a strong limitation with the onset of pain, while an extension and inclinations/rotations to the left allow the symptoms to cease.

The present pain could be connected to the mechanical tension generated by visceral adhesion, stimulated whenever the neck moved to the right. The nervous pathways put under tension are the same ones that innervate the pleural, pericardial area (vagus nerve and sympathetic system) [22]. Noxious information can derive from the respiratory viscera (stimulation of the P2X3 receptors) due to the presence of non-physiological mechanical tensions [23]. Similarly, altered mechanical tensions involving the pericardium (stimulating metabotropic glutamate receptors) can cause somato-visceral nociceptive afferents [24].

Muscle motor fibers traveling with the cervical and mediastinal fascia can be negatively solicited by mechanical tension abnormalities and form nociceptive information: movement with pain [22]. The phrenic nerve itself can carry, in an afferent way, nociceptive information for mechanical disturbances, where nociception can be expressed in distant muscle or joint areas of the neck [25]. It should be added that, in acute and chronic situations of non-specific neck pain, there is an alteration in the perception of local pain (mechanical hyperalgesia) with central sensitization, respectively [26]. After searching on PubMed for other methods of manual treatments, in particular, reviews with the terms "chronic non-specific neck pain physiotherapy", "chronic non-specific neck pain chiropractic", and "chronic non-specific neck pain manual therapy" for a possible comparison with the osteopathic approach, we did not find any methods considered to be gold standard [27–29]. A recent systematic review included 40 studies (randomized controlled trials, RCTs) to understand what the best rehabilitation exercise strategy would be for patients with NNP [27]. The authors concluded that, despite the large amount of research, there is no strategy that can be considered the gold standard, and there is a very low quality of evidence. A systematic review to evaluate the effectiveness of manual therapy for the approach to NNP took into consideration 23 RCTs; the results were very heterogeneous [28]. The difficulty in finding a more adequate manual strategy consists of the non-homogeneity of the manual technique

performed, and further studies are necessary to find a more effective manual therapy. A recent narrative review on chiropractic and NNP highlighted that not only does there not exist a gold standard of treatment, but the neurophysiological mechanisms underlying chiropractic techniques are not well understood.

It is not easy to identify the cause of NNP. Studies link the presence of this chronic disorder with other clinical pictures or behaviors, such as depression, anxiety, smoking, lack of regular physical activity, unrestful sleep, and previous traumas [30]. The etiology is heterogeneous, ranging from the presence of neurological diseases to structural or vascular alterations in the cervical vertebrae [30]. We can state that the lack of heterogeneity of the different research lies in the difficulty of homogenizing the patients and the cause (always subjective).

The osteopathic manual approach carried out in this case report finds its clinical validity in a previous trial, where we demonstrated that by working on the mediastinal area, the patient recovers their motor performance more quickly and with a significant decrease in pain perception [18]. The reasons that have made it possible to improve the clinical picture of the patients are not fully elucidated. The difference to the previous study is that in this clinical case, the position of the hands mirrored the position of the pericardial area. This is the first report that allows us to eliminate NNP, working the pericardial area with osteopathic treatment. It is probable that longer-term follow-up of the patient could lend further insights into the effectiveness and sustainability of this treatment method. The manual approach carried out does not cause side effects; in the literature, there are no texts that highlight dangers for the patient. The real reasons for the benefit achieved by the patient need to be investigated. We hope that this case report will stimulate further research.

## 4. Conclusions

This case report presents a patient with chronic non-specific neck pain (NNP) in the presence of pericardial-pleural adherence following thoracentesis and prior CABG. To the knowledge of the authors, this is the first article illustrating an osteopathic approach with resolution of the NNP, with a manual technique used on the pericardial area. The real reasons for the benefit achieved by the patient need to be investigated.

**Author Contributions:** All authors had the same role in drafting the article. All authors have read and agreed to the published version of the manuscript.

**Funding:** The article has been funded by the Italian Ministry of Health.

**Institutional Review Board Statement:** Ethics approval is not required for this case report.

**Informed Consent Statement:** Informed consent was obtained from the patient prior to the writing of this case report and included consent for the publication of case details.

**Data Availability Statement:** Being a case report, there is no further data.

**Conflicts of Interest:** The authors declare no conflict of interest.

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
