# Peer review of "Pericardial Adhesion and Chronic Non-Specific Neck Pain following Thoracentesis: An Osteopathic Approach"

_clinpract, doi:10.3390/clinpract13060117_

Round 1

Reviewer 1 Report

Comments and Suggestions for Authors

Line 16: "To the knowledge of the Authors, it is the first case report where

osteopathy is associated with a gentle treatment on the pericardial area to resolve a NNP." 

This statement is unclear in the abstract. Kindly rewrite to clarify.

Overall, a few inconsistencies in the manuscript need to be addressed. Although the authors have cited multiple manuscripts of their own, I feel they are apt and fit the literature presented well. I encourage these case reports; however, I believe that given the duration of the pain, "how much" the osteopathic approach truly helped the pain would be uncertain. But nonetheless, given the limited body of evidence on this topic, if this serves as a base for some patients in the future, it's worthy of a trial as long as no adverse effects or contraindications exist. I would like the authors to reiterate any contraindications, if any, to guide any future direction. 

Comments on the Quality of English Language

Editing is needed

Author Response

Dear Reviewer 1

Thanks for your time and advice.

As requested, we have highlighted that there are no known side effects in the literature with the approach presented in the case report. Further studies will be necessary to understand and improve any manual treatments in the cardiovascular surgical field.

We have changed the sentence in the abstract, as requested: “To the knowledge of the authors, it is the first article illustrating an osteopathic approach with resolution of NNP, with a manual technique on the pericardial area.”

We have added it at the end of the article, as requested: “The manual approach carried out does not cause side effects; in the literature there are no texts that highlight the dangers for the patient.” And “…; we hope that the case report will stimulate further research”.

Regarding the English language, we remember that Allan Escher is American. If the Editor decides to request a full revision in English, we will follow the Editor's advice.

Reviewer 2 Report

Comments and Suggestions for Authors

Thank you for the opportunity to review your article - "Osteopathic Approach in a Patient with Neck Pain Following CABG and Thoracentesis" - here is my assessment:

Strengths:

  1. Relevance: The topic is pertinent, as neck pain is a widespread clinical challenge and new methods of addressing it are of clinical interest.
  2. Comprehensive Introduction: The introduction effectively sets the context and importance of AMI and CABG. It adequately states the complications and problems arising from surgical interventions.
  3. Methodical Presentation: The detailed case presentation methodically walks through the patient's medical history, symptoms, differential diagnosis, and treatments, ensuring a clear understanding.
  4. Supporting References: The article is well-referenced, lending credibility to the claims and context provided.
  5. Use of Visual Aids: The inclusion of an anteroposterior chest x-ray provides a visual aid, which is beneficial for readers.

Areas for Improvement:

  1. Discussion Integration: The manuscript might benefit from a clearer integration of the discussion points with the presented case. The article sometimes tends to meander into hypothetical scenarios which may not be directly relevant to the core case.
  2. Patient Follow-up: Longer-term follow-up on the patient could lend further insights into the effectiveness and sustainability of the treatment method.
  3. Comparative Analysis: The discussion could benefit from a comparison of the osteopathic approach with other common treatment modalities for NNP, highlighting the unique advantages and possible limitations.
  4. Clarity on Novelty: The claim that this is the first article illustrating an osteopathic approach with resolution of NNP in this specific context should be further supported with a more exhaustive literature review.
  5. Broaden Discussion: Delving deeper into the physiological and biomechanical aspects of how adhesions after thoracentesis might affect neck pain could make the discussion richer.
Comments on the Quality of English Language

Review of the Manuscript: "Osteopathic Approach in a Patient with Neck Pain Following CABG and Thoracentesis"

Summary: The manuscript presents a case report of a patient with nonspecific neck pain (NNP) due to pericardial adherence after undergoing coronary artery bypass graft (CABG) and thoracentesis. The novelty lies in the osteopathic intervention used to address the patient's pain, which to the authors' knowledge, is the first report of its kind.

Strengths:

  1. Relevance: The topic is pertinent, as neck pain is a widespread clinical challenge and new methods of addressing it are of clinical interest.
  2. Comprehensive Introduction: The introduction effectively sets the context and importance of AMI and CABG. It adequately states the complications and problems arising from surgical interventions.
  3. Methodical Presentation: The detailed case presentation methodically walks through the patient's medical history, symptoms, differential diagnosis, and treatments, ensuring a clear understanding.
  4. Supporting References: The article is well-referenced, lending credibility to the claims and context provided.
  5. Use of Visual Aids: The inclusion of an anteroposterior chest x-ray provides a visual aid, which is beneficial for readers.

Areas for Improvement:

  1. Discussion Integration: The manuscript might benefit from a clearer integration of the discussion points with the presented case. The article sometimes tends to meander into hypothetical scenarios which may not be directly relevant to the core case.
  2. Patient Follow-up: Longer-term follow-up on the patient could lend further insights into the effectiveness and sustainability of the treatment method.
  3. Comparative Analysis: The discussion could benefit from a comparison of the osteopathic approach with other common treatment modalities for NNP, highlighting the unique advantages and possible limitations.
  4. Clarity on Novelty: The claim that this is the first article illustrating an osteopathic approach with resolution of NNP in this specific context should be further supported with a more exhaustive literature review.
  5. Broaden Discussion: Delving deeper into the physiological and biomechanical aspects of how adhesions after thoracentesis might affect neck pain could make the discussion richer.

Conclusion and Recommendation: The article offers a novel perspective on addressing neck pain post-CABG and thoracentesis using an osteopathic approach. While the manuscript is methodically laid out with a thorough case presentation, improvements can be made in streamlining the narrative and enriching the discussion.

Given the originality and clinical relevance of the topic, with necessary revisions, the article is worth publication. Future studies, perhaps with a larger patient cohort, would be valuable in further validating the presented osteopathic approach.

Author Response

Dear Reviewer 2

Thanks for your time and advice.

We have added the sentence in the "Discussion" section: "In the presence of adhesions, unfortunately, we do not have clear indications on the possible symptoms and we do not have a gold standard of manual treatment".

We have added the sentence in the "Discussion" section: “Probably, longer-term follow-up on the patient could lend further insights into the effectiveness and sustainability of the treatment method.”

We have added the sentence in the "Discussion" section: “Searching on PubMed for other methods of manual treatments, in particular, reviews with terms “chronic non-specific neck pain psysiotherapy”, “chronic non-specific neck pain chiropractic” and “chronic non-specific neck pain manual therapy” for a possible comparison with the osteopathic approach, we have not found any methods considered to be gold standard (27-29).” And we added references 27-29.

27 de Zoete RM, Armfield NR, McAuley JH, Chen K, Sterling M. Comparative effectiveness of physical exercise interventions for chronic non-specific neck pain: a systematic review with network meta-analysis of 40 randomised controlled trials. Br J Sports Med. 2020: bjsports-2020-102664. 10.1136/bjsports-2020-102664

28 Hidalgo B, Hall T, Bossert J, Dugeny A, Cagnie B, Pitance L. The efficacy of manual therapy and exercise for treating non-specific neck pain: A systematic review. J Back Musculoskelet Rehabil. 2017; 30(6):1149-1169. 10.3233/BMR-169615

29 Gevers-Montoro C, Provencher B, Descarreaux M, Ortega de Mues A, Piché M. Neurophysiological mechanisms of chiropractic spinal manipulation for spine pain. Eur J Pain. 2021; 25(7):1429-1448. 10.1002/ejp.1773

Reviewer 3 Report

Comments and Suggestions for Authors

Thank you very much for inviting me to review the case report entitled "Pericardial Adhesion and Chronic Nonspecific Neck Pain Following Thoracentesis: An Osteopathic Approach.”.

 The Authors presented a case where non-specific neck pain, in a patient undergoing CABG five years earlier. The described case is interesting, but requires improvement.

 Abstract:

 1.     This sentence is not clear:We present a case where non-specific neck pain (NNP), in a patient undergoing CABG five years earlier, was approached with an osteopathic technique, working the pericardial area”. Please correct this sentence.

 2.     ‘To the knowledge of the authors, it is the first article illustrating an osteopathic approach with resolution of NNP, with a manual technique on the pericardial area’. This sentence is more suitable for discussion. Please phrase your summary differently.

Introduction:

3.     ‘To the knowledge of the authors, it is the first article illustrating an osteopathic approach with resolution of NNP, with a manual technique on the pericardial area’. As above.

Case presentation:

4.     What does ‘double CABG’ mean? Please explain.

5.     Please expand the abbreviation ‘ACE’.

6.     ‘hypercholesterolemia and dyslipidemia’ mean one disease entity. Please select one diagnosis.

7.     Please expand the abbreviation ‘NSAID’.

8.     'the color Doppler ultrasound of the carotids previously performed by the patient'. This suggests that the patient was self-examining. Understands that the ultrasound was performed prior to the visit. Please change.

It is advisable to proofread the English language.

Comments on the Quality of English Language

It is advisable to proofread the English language.

Author Response

Dear Reviewer 3,

Thanks for your time and advice.

1 This sentence is not clear: ‘We present a case where non-specific neck pain (NNP), in a patient undergoing CABG five years earlier, was with an osteopathic technique, working the pericardial area”. Please correct this sentence.

We changed/added: “resolved”

2 ‘To the knowledge of the authors, it is the first article illustrating an osteopathic approach with resolution of NNP, with a manual technique on the pericardial area’. This sentence is more suitable for discussion. Please phrase your summary differently.

Answer: Dear Reviewer, we have changed in respect to Reviewer 1.

3 ‘To the knowledge of the authors, it is the first article illustrating an osteopathic approach with resolution of NNP, with a manual technique on the pericardial area’. As above.

We changed: “To the knowledge of the authors, it is the first article illustrating an osteopathic approach with resolution of NNP, with a manual technique on the pericardial area.”

4 What does ‘double CABG’ mean? Please explain.

Dear Reviewer, the meaning of the acronym was explained before, in the same section.

  1. Please expand the abbreviation ‘ACE’.

We Added: “angiotensin-converting enzyme”

  1. ‘hypercholesterolemia and dyslipidemia’ mean one disease entity. Please select one diagnosis.

Dear Reviewer, are two distinct clinical entities

  1. Please expand the abbreviation ‘NSAID’.

We added: non-steroidal anti-inflammatory drugs

  1. 'the color Doppler ultrasound of the carotids previously performed by the patient'. This suggests that the patient was self-examining. Understands that the ultrasound was performed prior to the visit. Please change.

We delete “by the patient”

It is advisable to proofread the English language.

Regarding the English language, we remember that Allan Escher is American. If the Editor decides to request a full revision in English, we will follow the Editor's advice.